# ATLAS: Adaptive Topology-based Learning at Scale for Homophilic and Heterophilic Graphs

## Abstract

We present ATLAS (Adaptive Topology-based Learning at Scale for Homophilic and Heterophilic Graphs), a novel graph learning algorithm that addresses two important challenges in graph neural networks (GNNs). First, the accuracy of GNNs degrades when the graph is heterophilic. Second, the iterative feature aggregation limits the scalability of GNNs on large graphs. We address these challenges by extracting topological information about the graph communities at different levels of refinement, concatenating the community assignments to the feature vector, and applying multilayer perceptrons (MLPs) on this new feature vector. We thus inherently obtain the topological data about the nodes and their neighbors without invoking aggregation. Because MLPs are typically more scalable than GNNs, our approach applies to large graphs, without the need for sampling.

Our results, on a wide set of graphs, show that ATLAS has comparable accuracy to baseline methods, with accuracy being as high as 20 percentage points over GCN for heterophilic graphs with negative structural bias and 11 percentage points over MLP for homophilic graphs. Furthermore, we show how multi-resolution community features systematically modulate performance in both homophilic and heterophilic settings, opening a principled path toward explainable graph learning.

## 1 Introduction

Node classification, a fundamental problem in graph learning, involves identifying labels of nodes in a graph and has wide applications in many domains including social networks, citation networks, recommendation systems, knowledge graphs and bioinformatics (Khemani, 2024; Wu et al., 2019b; Zhou et al., 2021). Accurate classification requires two complementary pieces of information–(i) the features at each node, and (ii) the connections between the node and its neighbors. Neural network methods such as Multi-Layer Perceptrons (MLPs) are fast but do not include information about the connections. Graph Neural Networks (GNNs) address this problem by aggregating the features between neighboring nodes, but the process is expensive, and difficult to scale to large graphs. Although the graph structure can be represented as feature vectors using different node embedding techniques (Perozzi et al., 2014; Grover & Leskovec, 2016; Tang et al., 2015), or through the use of community detection (Sun et al., 2019; Kamiński et al., 2024), the issue remains as to how many hops of neighbors should be considered and how fine-grained the communities should be. Larger hops or coarse grained community can lead to information smoothing, while smaller hops or fine grained communities can lead to information loss. Further, the hypothesis that aggregating features of neighbors can improve accuracy of node classification is only true for homophilic networks (where nodes of similar classes are connected). In heterophilic networks, where the connection between nodes need not imply similarity of class, this strategy leads to lower accuracy. Based on these observations, we posit, *matching structural information (i.e. size of hops or communities) with how well it aligns with the classification is necessary for producing accurate results.*

### 1.1 Related Work

Graph Neural Networks (GNNs) have become a core tool for learning on graphs (Kipf & Welling, 2017; Hamilton et al., 2017). Most algorithms follow a message-passing paradigm, aggregating transformed neighbor features into topology-aware embeddings, which implicitly assumes *homophily*

(Wu et al., 2019a). The same bias can blur informative distinctions on weakly homophilous or heterophilous graphs (Zhu et al., 2020; Platonov et al., 2023a).

**Scaling GNNs on large graphs.** Scaling GNNs on large graphs is challenging due to memory and aggregation costs. Sampling-based methods approximate full-batch propagation using node-, layer-, or subgraph-level sampling (GraphSAGE, FastGCN, Cluster-GCN, GraphSAINT, LABOR) (Hamilton et al., 2017; Chen et al., 2018b; Chiang et al., 2019; Zeng et al., 2020; Balın & Çatalyürek, 2023), but introduce stochasticity that affects convergence and reproducibility (Chen et al., 2018b; Zou et al., 2019). Decoupled models instead precompute feature diffusion and train MLPs on fixed graph-derived features, enabling i.i.d. node mini-batching and fast inference (SGC, SIGN, SAGN, GAMLP, SCARA, LD$^2$) (Wu et al., 2019a; Rossi et al., 2020; Sun et al., 2021; Chien et al., 2022; Liao et al., 2022; 2023).

**Learning on non-homophilous graphs.** For non-homophilous graphs, one line of work preserves self-features while carefully injecting neighborhood information (H2GCN, GloGNN) (Zhu et al., 2020; Li et al., 2022), or reweights neighbors to downweight harmful edges (GPR-GNN, FAGCN) (Chien et al., 2021; Bo et al., 2021). Others exploit higher-order propagation or spectral filters to capture both homophilic and heterophilic signals (MixHop, JacobiConv, BernNet, GBK-GNN) (Abu-El-Haija et al., 2019; Wang & Zhang, 2022; He et al., 2021; Du et al., 2022). See Zheng et al. (2022); Luan et al. (2024b) for broader surveys.

**Community-aware node embeddings.** Several works use community structure as an explicit representation for downstream prediction. Sun et al. (2019) propose vGraph, a generative model that jointly infers discrete communities and continuous node embeddings by reconstructing edges, so that community assignments act as latent variables guiding representation learning. Closer to our setting, Kamiński et al. (2024) construct community-aware node features (e.g., counts and statistics over community memberships in a node's ego-network) and feed them into standard classifiers, showing that purely community-derived signals can already yield strong performance on node-level tasks.

**Graph–task alignment and community structure.** A related line of work asks when a graph's communities are informative for the labels and how this alignment controls the benefit of message passing. Hussain et al. (2021) vary homophily and community structure in real graphs and define a measure of label–community correlation, showing that GNN gains are largest when labels follow communities and can vanish when they do not. This links the classical "cluster assumption" in semi-supervised learning (Chapelle et al., 2006) with recent analyses of graph–task and NTK–graph alignment in GNN training dynamics (Yang et al., 2024), and motivates methods that treat communities as task-relevant structural signals.

**Community-guided graph rewiring.** Building on modularity-based detection (Newman, 2006; Blondel et al., 2008), ComMa and ComFy (Rubio-Madrigal et al., 2025) use community structure and feature similarity to rewire intra- and inter-community edges, improving label–community alignment and GNN accuracy on both homophilic and heterophilic graphs.

Unlike community-aware GNNs and rewiring methods, ATLAS treats multi-resolution community assignments as features for a simple MLP, remaining propagation-free while still leveraging community structure.

## 1.2 OUR CONTRIBUTION

Most of the current research either focuses primarily on homophilic graphs, or the processes to address the heterophilic graphs require expensive operations, such as signal identification/modification, rewiring or spectral gap maximization. These methods cannot efficiently scale to large graphs. Our **primary contribution** is to develop **A**daptive **T**opology -based **L**earning **a**t **S**cale (**ATLAS**) *, a novel graph learning algorithm that can produce high-accuracy results for both homophilic and heterophilic graphs. ATLAS is based on a *simple but powerful technique of refining communities in networks to match the degree of homophily.*

**Rationale.** Our algorithm is based on quantifying homophily through the lens of normalized mutual information (NMI). Given two partitions of the same set of elements NMI measures how well the

---

*Apart from the acronym, the name ATLAS is to convey our method can handle different degrees of homophily, similar to how an atlas encompasses all different countries.

partitions correspond to each other. If we consider one partition as the communities in the graph, and the other partition as labels, then NMI provides a measure for the degree of homophily in the graph. ATLAS focuses on refining/coarsening communities to identify the region of highest NMI—which will correspond to the highest accuracy. Figure 1 provides an overview of ATLAS.

Our specific contributions are:

1. **Theory.** We provide a theoretical analysis of how refining communities changes in NMI (Section 2).

2. **Algorithm.** Based on this mathematical understanding, we develop our algorithm ATLAS (Section 3).

3. **Experiments.** Provide extensive empirical evaluations by comparing ATLAS across a mix of 13 (8 medium size and 5 large) homophilic and heterophilic graphs, and 14 (9) GNN/MLP-based algorithms for medium sized (large) graphs (Section 4).

4. **Bridging Frameworks.** Unlike prior MLP-based models designed primarily for heterophilic graphs, ATLAS effectively supports both homophilic and heterophilic settings, thereby minimizing the accuracy gap traditionally observed between MLPs and GNNs. Moreover, its high inference efficiency positions ATLAS as a practical and scalable alternative to GNNs.

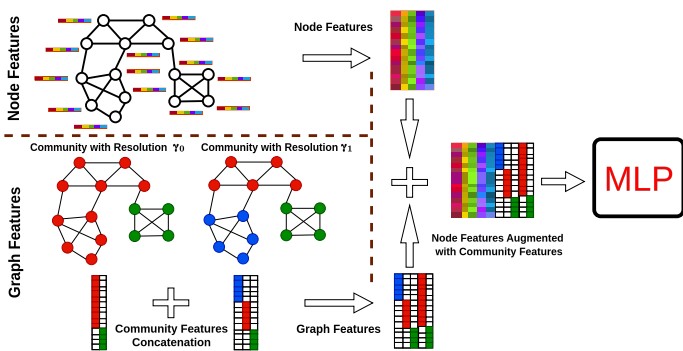

Figure 1: Overview of the community-augmented feature learning pipeline. Community assignments at multiple resolutions are one-hot encoded, projected, concatenated with node features, and input to an MLP for classification.

## 2 THEORETICAL ANALYSIS

We mathematically show how refining communities leads to changes in NMI. We define some terms that will help us in the analysis. The proofs of the theorems are given in the appendix.

Let $N$ be the set of nodes. Let $P = \{P_1, \ldots, P_k\}$ be a partition of $N$; i.e.

$$P_i \neq \varnothing, \quad P_i \cap P_j = \varnothing \; (i \neq j), \quad \text{and} \quad \bigcup_{i=1}^{k} P_i = N.$$

Let $S = \{S_1, \ldots, S_m\}$ be another partition of $N$. We say $S$ *is a refinement of* $P$ (denoted as $S \preceq P$) iff every block of $S$ is contained in some block of $P$. Formally:

$$S \preceq P \quad \Longleftrightarrow \quad \forall S_j \in S \; \exists P_i \in P \text{ such that } S_j \subseteq P_i.$$

*Normalized mutual information (NMI)* is a popular measure to quantify alignment between two partitions. Given two partitions $P$ and $Q$, over a set of $N$ elements and $n_{ij} = |P_i \cap Q_j|, n_i = |P_i|, n_j = |Q_j|$ their normalized mutual information is given as;

$$\text{NMI}(P, Q) = \frac{2I(P;Q)}{H(P) + H(Q)}$$

$I(P;Q) = \sum_{i=1}^{k} \sum_{j=1}^{m} \frac{n_{ij}}{N} \log \left( \frac{N \, n_{ij}}{n_i \, n_j} \right)$ is the mutual information between partitions $P$ and $Q$. This quantity measures how much information is shared between the partitions $P$ and $Q$. The higher

the value, the better the alignment between the partitions. $H(P) = -\sum_{i=1}^{k} \frac{n_i}{N} \log\left(\frac{n_i}{N}\right)$, is the entropy of partition $P$. $H(Q)$ is defined similarly. The entropy measures the distribution of points in each partition. Low entropy means data is concentrated in few clusters, and is indicative of good clustering.

The value of NMI ranges from 1 (indicating complete alignment between partitions) to close to 0 (indicating complete mismatch between partitions). NMI is high if the partitions are well matched ($I(P, Q)$ is high), and entropy is low ($H(P)$, $H(Q)$ is low).

**Lemma 1** (Refinement does not decrease mutual information). *Let $L$ be labels and $C$ a community partition. Let $C'$ be a refinement of $C$, i.e., $C' \preceq C$. Then $I(L; C') \geq I(L; C)$*

**Lemma 2** (Refinement does not decrease entropy). *Let $C$ a community partition. Let $C'$ be a refinement of $C$, i.e., $C' \preceq C$. Then $H(C') \geq H(C)$*

Based on Lemma 1 and Lemma 2 we see that while refinement improves the mutual information leading to better alignment, it also increases the entropy leading to more noise or uncertainty. The condition at which NMI will increase is given by Theorem 1.

**Theorem 1** (NMI Refinement Condition). *Let $L$ be labels; $C$ a community partition. Let $C'$ be a refinement of $C$, i.e., $C' \preceq C$. Then $NMI(C'; L) > NMI(C; L)$ if and only if $\frac{\Delta I}{\Delta H} > \frac{NMI(C; L)}{2}$; where $\Delta I = I(C'; L) - I(C; L)$ and $\Delta H = H(C'; L) - H(C; L)$*

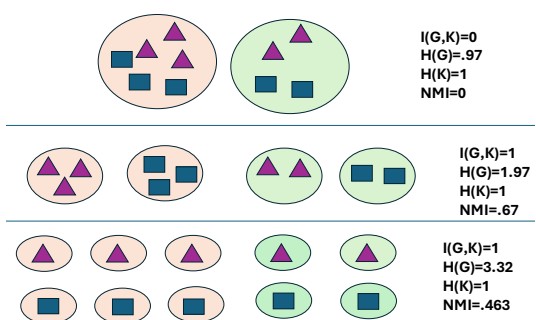

Figure 2: Effect of refinement on NMI. Initially when clusters have mixed items, NMI is low. The first refinement matches the items and clusters, increasing the NMI. Further refinement does not improve the alignment (mutual information), but increases the spread (entropy), thus decreasing NMI.

Theorem 1 states that a partition refinement improves the normalized mutual information with respect to labels if and only if the mutual information gain per unit of entropy increase exceeds half the original normalized mutual information value.

## 3 METHODOLOGY

The theorems in Section 2 are based on idealized conditions, where refined communities are perfect subsets of the original communities. In practice, refinement in communities is approximated by running a modularity-based community detection algorithm at multiple resolution values. Although higher resolution leads to smaller communities, due to the inherent non-determinism of community detection methods, the smaller communities may not be exact subsets.

**Preprocessing.** Optimizing modularity is a popular method for community detection. Modularity, $Q$, measures the strength of connections between nodes in a community as compared to a null model with randomly placed edges. Communities in networks are often hierarchical, so we treat the resolution parameter $\gamma$ as the hierarchy/refinement level (larger $\gamma$ yields finer-grained communities); Appendix 8.1 summarizes the community terminology and formal definitions used below. We start from two initial resolutions ($\gamma = 0.5$ and $\gamma = 1.0$) and set three hyperparameters: a modularity gap threshold $\Delta_{\max}$, a minimum modularity $Q_{\min}$, and a small target-drop range $[a, b]$. Let $\gamma_1$ and $\gamma_2$ be two consecutive resolution parameters, with community sets $\mathbf{c}^{(\gamma_1)}$ and $\mathbf{c}^{(\gamma_2)}$ and modularities $Q^{(\gamma_1)}$ and $Q^{(\gamma_2)}$; we define the modularity gap as $\Delta Q = |Q^{(\gamma_2)} - Q^{(\gamma_1)}|$. At each iteration, we sort the tested resolutions and examine consecutive pairs. If the gap between a pair exceeds $\Delta_{\max}$, we find their midpoint (interpolation). Otherwise, we extrapolate beyond the current maximum by estimating the local slope of modularity with respect to the resolution and taking a small forward step expected to reduce modularity by a random amount drawn from the drop range. Once the new $\gamma$ is obtained, we compute the communities at that value. The loop stops when the latest modularity falls below

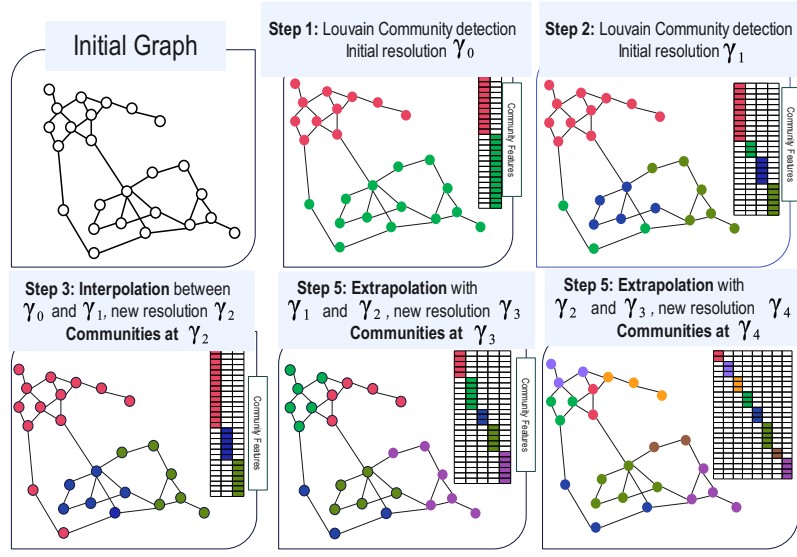

Figure 3: Illustration of the Adaptive Resolution Search Process. The resolution limits, $\gamma_0 < \gamma_2 < \gamma_1 < \gamma_3 < \gamma_4$, and the communities $C^{\gamma_0} \preceq C^{\gamma_2} \preceq C^{\gamma_1} \preceq C^{\gamma_3} \preceq C^{\gamma_4}$ capture structural bias for different granularities from the graph.

$Q_{\min}$ or no new resolution is produced. The procedure returns the retained resolutions and their corresponding community assignments, which we view as multi-resolution community features—a structural graph signal over the nodes—that are later encoded and concatenated with the original node features in the feature-augmentation step (see Algorithm 1 in the appendix).

**Feature Augmentation.** For a given resolution parameter $\gamma$, let the communities be $\mathbf{c}^{(\gamma)} \in \{1, \ldots, k_\gamma\}$, and and each node is assigned to one of the communities in $k_\gamma$. This assignment is represented as a one-hot encoded matrix $\mathbf{H}^{(\gamma)}$( equation 3). To reduce dimensionality, each one-hot matrix is projected into a dense embedding space using a trainable weight matrix $\mathbf{W}^{(\gamma)}$ ( equation 4). The embeddings from all resolutions are concatenated to form $\mathbf{E}$ ( equation 5), which is then further concatenated with the original features $\mathbf{X}$ to yield the augmented feature matrix $\mathbf{Z}$. The augmented feature matrix $\mathbf{Z}$ is fed to an MLP $f_\theta$ to produce logits; a task-dependent function $\phi$ (e.g., softmax or elementwise sigmoid), applied row-wise, converts them to probabilities $\hat{\mathbf{Y}}$ ( equation 7).

.

$$\mathbf{X} \in \mathbb{R}^{n \times D}, \quad \Gamma = \{\gamma_1, \ldots, \gamma_T\} \tag{1}$$

$$\mathbf{c}^{(\gamma)} = \text{DetectCommunity}(G, \gamma), \quad \mathbf{c}^{(\gamma)} \in \{1, \ldots, k_\gamma\}^n \tag{2}$$

$$\mathbf{H}^{(\gamma)} = \text{OneHot}(\mathbf{c}^{(\gamma)}) \in \{0, 1\}^{n \times k_\gamma} \tag{3}$$

$$\mathbf{E}^{(\gamma)} = \mathbf{H}^{(\gamma)}\mathbf{W}^{(\gamma)}, \quad \mathbf{W}^{(\gamma)} \in \mathbb{R}^{k_\gamma \times d_c} \tag{4}$$

$$\mathbf{E} = \big\|_{t=1}^{T} \mathbf{E}^{(\gamma_t)} \in \mathbb{R}^{n \times (Td_c)} \tag{5}$$

$$\mathbf{Z} = [\mathbf{X} \parallel \mathbf{E}] \in \mathbb{R}^{n \times (D + Td_c)} \tag{6}$$

$$\hat{\mathbf{Y}} = \phi\big(f_\theta(\mathbf{Z})\big) \in [0, 1]^{n \times C} \tag{7}$$

**Complexity Analysis.** We compare computational and memory complexities of representative scalable GNN frameworks with our approach in Table 3. ATLAS performs Louvain clustering in $O(T\|A\|_0)$ in the preprocessing step, keeps a single augmented feature buffer, and trains with per-epoch time $O\big(L_{ff}N(D+Td_c)^2\big)$ and memory $O\big(bL_{ff}(D+Td_c)\big)$, enabling simple i.i.d. node mini-batching without neighborhood expansion or graph-dependent batching heuristics. ATLAS

performs adjacency-free inference: with fixed augmented features of dimension $D+Td_c$, prediction is a forward pass with complexity $O\big(N(D+Td_c)^2\big)$.

## 4 EMPIRICAL EVALUATION

In this section, we provide the empirical results comparing ATLAS with other graph learning methods. Our experiments focus on answering the following *research questions*:

**Q1.** How accurate is ATLAS compared to baseline methods over graphs with different degrees of homophily?

**Q2.** How well can ATLAS scale to large graphs, while maintaining high accuracy?

**Datasets.** We use 8 medium graphs (Cora, PubMed, Tolokers, Squirrel-Filtered, Chameleon-Filtered, Amazon-Ratings, Actor, Roman-Empire) and 5 large graphs (Flickr, Reddit, Yelp, Amazon-Products, OGBN-Products). Complete statistics of datasets are given in Appendix Tables 8 and 9.

**Baselines.** We group baselines by modeling regime and map them to the research questions.

*Q1 (homophily–heterophily regime). Homophilic*: GCN (Kipf & Welling, 2017), GraphSAGE (Hamilton et al., 2017), GAT (Veličković et al., 2018). *Heterophily-oriented*: H$_2$GCN (Zhu et al., 2020), LinkX (Lim et al., 2021), GPR-GNN (Chien et al., 2021), FSGNN (Maurya et al., 2022), GloGNN (Li et al., 2022), FAGCN (Bo et al., 2021), GBK-GNN (Du et al., 2022), JacobiConv (Wang & Zhang, 2022), ACM-GCN (Luan et al., 2022), BernNet (He et al., 2021).

*Q2 (scalability). Propagation-free / decoupled*: SGC (Wu et al., 2019a), SIGN (Rossi et al., 2020), SAGN (Sun et al., 2021), GAMLP (Chien et al., 2022). *Sampling-based*: GraphSAGE (Hamilton et al., 2017), ClusterGCN (Chiang et al., 2019), GraphSAINT (Zeng et al., 2020). Descriptions of these methods are provided in the Appendix.

We use an $L$-layer MLP with hidden width $d_{\text{hid}}$ and dropout rate $p$. Each of the first $L-1$ layers applies *Linear (with bias) $\rightarrow$ LayerNorm $\rightarrow$ GELU $\rightarrow$ Dropout*. The final layer is a *Linear* classifier to $C$ classes.

### 4.1 Q1: ACCURACY ACROSS HOMOPHILY REGIMES

Table 1 reports results on the eight medium-sized benchmarks. We group these datasets into three *structural-bias* regimes—high, low, and negative structural bias—based on how informative their community structure is for the labels; we formalize this notion in Section 5. On graphs with negative structural bias, ATLAS improves over GCN by up to 20 percentage points, and on high structural-bias graphs it improves over a feature-only MLP by more than 11 percentage points. Although it does not attain the best score on every dataset, an appropriate choice of resolution parameters typically allows ATLAS to match or closely track the strongest baseline across both homophilic and heterophilic regimes. The main outlier is Roman-Empire, where accuracy appears to be largely driven by raw node features. FSGNN explicitly concatenates neighborhood features to strengthen this input signal. When we equip ATLAS with the same neighborhood-feature concatenation (ATLAS-NF), its accuracy on Roman-Empire rises to within roughly two percentage points of FSGNN, while also achieving the best performance on Tolokers.

Overall, ATLAS and its neighbor-feature variant ATLAS-NF substantially narrow the MLP$\rightarrow$GNN performance gap and provide a single topology-augmented architecture that is consistently competitive with the strongest model on each dataset, across all three structural-bias regimes.

### 4.2 Q2: EFFICIENCY AND SCALABILITY ON LARGE GRAPHS

**Accuracy.** Table 2 shows that ATLAS scales to million-node graphs and is competitive across all structural-bias regimes while consistently improving over MLP and GCN. On high structural-bias graphs, it stays close to the best methods with gains up to about +0.21 over MLP and +0.03 over GCN. On Flickr, ATLAS-NF (ATLAS with neighbor features) surpasses ATLAS, indicating the impact of enhanced feature signal from neighbours, and on negative structural-bias graphs ATLAS maintains strong performance while aggregation-heavy GNNs degrade.

Table 1: Eight-benchmark comparison across homophily regimes. Baseline heterophily-oriented model results are from Platonov et al. (2023b); Luan et al. (2024a). Bottom rows report ATLAS improvements over baselines (absolute percentage points). Cells highlighted in yellow indicate the best score for each dataset.

| Model | High structural bias | | Low structural bias | | | Negative structural bias | | |
|---|---|---|---|---|---|---|---|---|
| | Cora $h_e = 0.810$ | Tolokers $h_e = 0.595$ | PubMed $h_e = 0.802$ | Chameleon-Filtered $h_e = 0.236$ | Amazon-Ratings $h_e = 0.380$ | Actor $h_e = 0.216$ | Squirrel-Filtered $h_e = 0.207$ | Roman-Empire $h_e = 0.047$ |
| MLP(2L) | 75.44 ± 1.97 | 72.97 ± 0.90 | 87.25 ± 0.41 | 36.00 ± 4.69 | 39.83 ± 0.48 | 34.96 ± 0.71 | 34.29 ± 3.34 | 65.58 ± 0.34 |
| GCN | 87.01 ± 1.04 | 74.93 ± 1.32 | 86.71 ± 0.42 | 37.11 ± 3.04 | 42.78 ± 0.14 | 28.49 ± 0.91 | 32.70 ± 1.73 | 45.68 ± 0.38 |
| SAGE | 87.50 ± 0.87 | 80.95 ± 0.92 | 88.42 ± 0.55 | 38.83 ± 4.26 | 44.67 ± 0.51 | 34.08 ± 1.07 | 33.32 ± 1.75 | 76.21 ± 0.65 |
| GAT | 87.74 ± 0.88 | 75.31 ± 1.35 | 86.18 ± 0.64 | 37.18 ± 3.44 | 43.25 ± 0.85 | 29.11 ± 1.23 | 32.61 ± 2.06 | 47.16 ± 0.66 |
| H2GCN | 87.52 ± 0.61 | 73.35 ± 1.01 | 87.78 ± 0.28 | 26.75 ± 3.64 | 36.47 ± 0.23 | 38.85 ± 1.17 | 35.10 ± 1.15 | 60.11 ± 0.52 |
| LinkX | 82.62 ± 1.44 | 81.15 ± 1.23 | 88.12 ± 0.47 | 40.10 ± 2.21 | 52.66 ± 0.64 | 35.64 ± 1.36 | 42.34 ± 4.13 | 56.15 ± 0.93 |
| GPR-GNN | 79.51 ± 0.36 | 72.94 ± 0.97 | 85.07 ± 0.09 | 39.93 ± 3.30 | 44.88 ± 0.34 | 39.30 ± 0.27 | 38.95 ± 1.99 | 64.85 ± 0.27 |
| FSGNN | 87.51 ± 1.21 | 82.76 ± 0.61 | 90.11 ± 0.43 | 40.61 ± 2.97 | 52.74 ± 0.83 | 37.65 ± 0.79 | 35.92 ± 1.32 | 79.92 ± 0.56 |
| GloGNN | 87.67 ± 1.16 | 73.39 ± 1.17 | 90.32 ± 0.54 | 25.90 ± 3.58 | 36.89 ± 0.14 | 39.65 ± 1.03 | 35.11 ± 1.24 | 59.63 ± 0.69 |
| FAGCN | 88.85 ± 1.36 | 77.75 ± 1.05 | 89.98 ± 0.54 | 41.90 ± 2.72 | 44.12 ± 0.30 | 31.59 ± 1.37 | 41.08 ± 2.27 | 65.22 ± 0.56 |
| GBK-GNN | 87.09 ± 1.52 | 81.01 ± 0.67 | 88.88 ± 0.44 | 39.61 ± 2.60 | 45.98 ± 0.71 | 38.47 ± 1.53 | 35.51 ± 1.65 | 74.57 ± 0.47 |
| JacobiConv | 89.61 ± 0.96 | 68.66 ± 0.65 | 89.99 ± 0.39 | 39.00 ± 4.20 | 43.55 ± 0.48 | 37.48 ± 0.76 | 29.71 ± 1.66 | 71.14 ± 0.42 |
| BernNet | 88.52 ± 0.95 | 77.00 ± 0.65 | 88.48 ± 0.41 | 40.90 ± 4.06 | 44.64 ± 0.56 | 41.79 ± 1.01 | 41.18 ± 1.77 | 65.56 ± 1.34 |
| ACM-GCN | 89.75 ± 1.16 | 74.95 ± 1.16 | 90.96 ± 0.62 | 42.73 ± 3.59 | 52.49 ± 0.24 | 41.86 ± 1.48 | 42.35 ± 1.97 | 71.89 ± 0.61 |
| ATLAS | 87.09 ± 1.62 | 82.19 ± 0.73 | 88.85 ± 0.48 | 42.76 ± 3.47 | 53.15 ± 0.61 | 38.48 ± 0.93 | 40.35 ± 1.53 | 66.22 ± 0.53 |
| ATLAS-NF | 86.73 ± 1.04 | 83.02 ± 0.74 | 88.76 ± 0.36 | 40.02 ± 2.79 | 52.30 ± 0.64 | 34.26 ± 0.97 | 36.98 ± 2.37 | 77.94 ± 0.48 |
| ATLAS–MLP (pp) | +11.65 | +9.22 | +1.60 | +6.76 | +13.32 | +3.52 | +6.06 | +0.64 |
| ATLAS–GCN (pp) | +0.08 | +7.26 | +2.14 | +5.65 | +10.37 | +9.99 | +7.65 | +20.54 |
| ATLAS–Average (pp) | +0.92 | +5.97 | +0.40 | +5.15 | +8.51 | +2.13 | +3.91 | +1.67 |

Table 2: Large-graph performance. Baselines from Zeng et al. (2020); Hu et al. (2020). Bottom rows report ATLAS improvements over MLP and over GCN (absolute units). Cells highlighted in yellow indicate the best score for each dataset.

| Method | High structural bias | | Low structural bias | Negative structural bias | |
|---|---|---|---|---|---|
| | Reddit $h_e=0.756$ | ogbn-products $h_e=0.808$ | Flickr $h_e=0.319$ | Yelp $h_e=0.809$ | AmazonProducts $h_e=0.116$ |
| MLP | 0.7435 ± 0.0016 | 0.6106 ± 0.0008 | 0.4717 ± 0.0011 | 0.6546 ± 0.0011 | 0.8204 ± 0.0002 |
| GCN | 0.9330 ± 0.0001 | 0.7564 ± 0.0021 | 0.4920 ± 0.0030 | 0.3780 ± 0.0010 | 0.2810 ± 0.0050 |
| GraphSAGE | 0.9530 ± 0.0010 | 0.8061 ± 0.0016 | 0.5010 ± 0.0130 | 0.6340 ± 0.0060 | 0.7580 ± 0.0020 |
| ClusterGCN | 0.9540 ± 0.0010 | 0.7862 ± 0.0061 | 0.4810 ± 0.0050 | 0.6090 ± 0.0050 | 0.7590 ± 0.0080 |
| GraphSAINT | 0.9660 ± 0.0010 | 0.7536 ± 0.0034 | 0.5110 ± 0.0010 | 0.6530 ± 0.0030 | 0.8150 ± 0.0010 |
| SGC | 0.9351 ± 0.0004 | 0.6748 ± 0.0011 | 0.5035 ± 0.0005 | 0.2356 ± 0.0002 | 0.2262 ± 0.0028 |
| SIGN | 0.9595 ± 0.0002 | 0.8052 ± 0.0016 | 0.5160 ± 0.0011 | 0.5798 ± 0.0012 | 0.7424 ± 0.0002 |
| SAGN | 0.9648 ± 0.0003 | 0.8121 ± 0.0007 | 0.5007 ± 0.0011 | 0.6155 ± 0.0040 | 0.7682 ± 0.0115 |
| GAMLP | 0.9673 ± 0.0003 | 0.8376 ± 0.0019 | 0.5258 ± 0.0012 | 0.5784 ± 0.0154 | 0.7599 ± 0.0026 |
| ATLAS | 0.9574 ± 0.0004 | 0.7865 ± 0.0053 | 0.5104 ± 0.0039 | 0.6546 ± 0.0011 | 0.8204 ± 0.0002 |
| ATLAS-NF | 0.9410 ± 0.0007 | 0.7507 ± 0.0030 | 0.5201 ± 0.0012 | 0.5740 ± 0.0021 | 0.7783 ± 0.0010 |
| ATLAS–MLP | +0.2139 | +0.1759 | +0.0387 | +0.0000 | +0.0000 |
| ATLAS–GCN | +0.0244 | +0.0301 | +0.0184 | +0.2766 | +0.5394 |
| ATLAS–Average | +0.0267 | +0.0262 | +0.0101 | +0.1059 | +0.1615 |

**Convergence.** ATLAS converges rapidly and stably across large graphs: training loss decreases smoothly, and validation performance plateaus early with a small train–validation gap. The curves exhibit no late-epoch degradation and remain stable after convergence (see Fig. 4).

**Efficiency.** Table 3 shows that ATLAS adds a $T$-resolution community search as a one-time preprocessing step with cost $O(T\|A\|_0)$, after which training is MLP-like and inference is adjacency-free on features of dimension $D + Td_c$. Consequently, both preprocessing and inference costs increase with $T$. On OGBN-Products (Table 4), the larger $T$ leads to a noticeable preprocessing cost, yet per-epoch training remains competitive and inference stays sub-second. Appendix Table 5 reports timings on the remaining large graphs.

Table 3: Complexity comparison. $N$ = #nodes, $\|A\|_0$ = #edges, $D$ = feature dim, $L$ = #message-passing layers, $L_{ff}$ = #feed-forward layers, $b$ = batch size, $r$ = sampled neighbors (or filter size), $K$ = #precomputed hop propagations (max hop order for SAGN/GAMLP), $k$ = #subgraph samples used in GraphSAINT preprocessing, $T$ = #resolutions, $d_c$ = community-embedding dim.

| Method | Preprocessing | Per-epoch Train Time | Memory |
|---|---|---|---|
| GCN (full-batch) | – | $O(L\|A\|_0 D + LND^2)$ | $O(LND + LD^2)$ |
| ClusterGCN | $O(\|A\|_0)$ | $O(L\|A\|_0 D + LND^2)$ | $O(bLD + LD^2)$ |
| GraphSAINT | $O(kN)$ | $O(L\|A\|_0 D + LND^2)$ | $O(bLD)$ |
| SAGN | $O(K\|A\|_0 D)$ | $O(L_{ff}N(KD)^2)$ | $O(bL_{ff}KD)$ |
| GAMLP | $O(K\|A\|_0 D)$ | $O(L_{ff}N(KD)^2)$ | $O(bL_{ff}KD)$ |
| **ATLAS** | $O(T\|A\|_0)$ | $O(L_{ff}N(D + Td_c)^2)$ | $O(bL_{ff}(D + Td_c))$ |

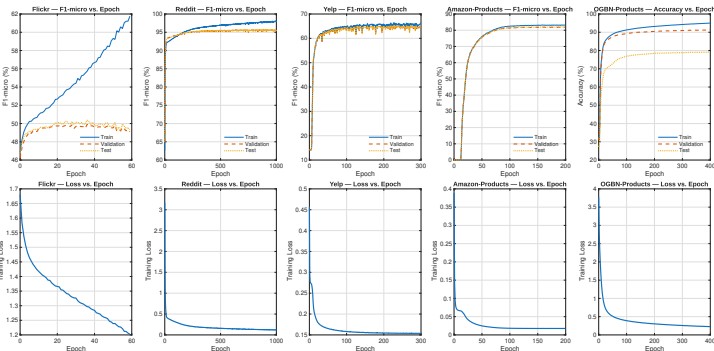

Figure 4: The convergence landscape of ATLAS.

Table 4: Computation time breakdown (seconds) on OGBN-Products.

| Model | Preprocessing Time | Per-epoch Train Time | Inference Time |
|---|---|---|---|
| GCN | — | $2.0395 \pm 0.0006$ | $0.9220 \pm 0.0010$ |
| ClusterGCN | $168.754 \pm 1.777$ | $4.017 \pm 0.164$ | $82.837 \pm 0.622$ |
| GraphSAINT | $3.770 \pm 0.159$ (per epoch) | $0.751 \pm 0.046$ | $66.445 \pm 0.517$ |
| SAGN | $4.8462 \pm 0.0415$ | $0.8447 \pm 0.0225$ | $0.2564 \pm 0.0001$ |
| GAMLP | $4.8227 \pm 0.0871$ | $0.7976 \pm 0.0099$ | $0.2495 \pm 0.0001$ |
| **ATLAS** | $391.894 \pm 14.387$ | $0.181 \pm 0.0058$ | $0.526 \pm 0.0038$ |

## 5 ACCURACY UNDER COMMUNITY REFINEMENT

We quantify the level of community refinement by the minimum modularity threshold $Q_{\min}$. Large $Q_{\min}$ preserves only coarse communities; lowering $Q_{\min}$ progressively adds medium and fine communities, yielding a multi-scale representation. We define *structural bias* as how strongly a graph's community structure provides a useful structural signal for classification, and group graphs into three regimes:

- **High structural bias** (e.g., Cora, Tolokers): Community structure is strongly aligned with labels, so refinement helps. Coarse communities at large $Q_{\min}$ already carry substantial signal, and adding medium- and fine-grained communities reveals additional useful structure. As $Q_{\min}$ decreases and more refined communities are included, performance steadily improves until it saturates; see Figure 5 (top left and bottom left). In this regime, GCN and ATLAS both outperform MLP.

- **Low structural bias** (e.g., Amazon-Ratings, Chameleon-Filtered, Flickr): Community structure is only weakly label-aligned. Coarse communities capture most of this limited structural signal, and adding finer communities yields at most small additional gains. As $Q_{\min}$ decreases and more refined communities are included, ATLAS improves moderately over the MLP, while GCN shows at best small or sometimes no gains over MLP; see Figure 5 (top center and bottom center). Here, topology provides some extra information, but node features remain the primary driver of performance.

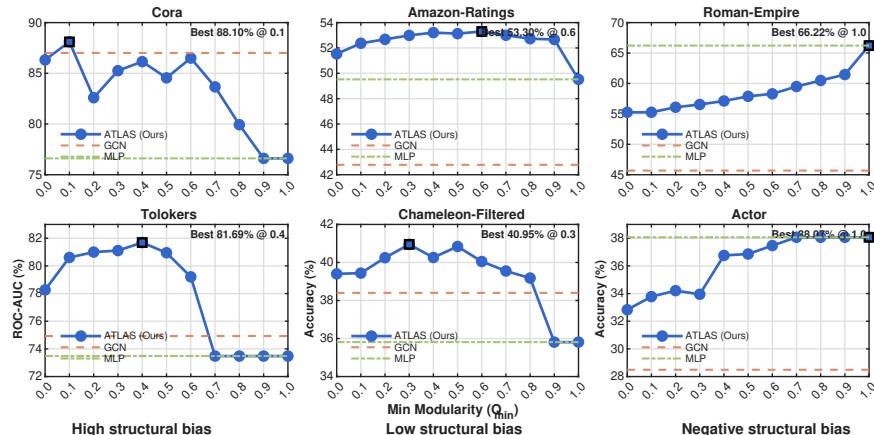

Figure 5: Effect of cumulatively adding community-derived features as the minimum modularity threshold $Q_{\min}$ is lowered, for high structural bias graphs (left), low structural bias graphs (middle), and negative structural bias graphs (right).

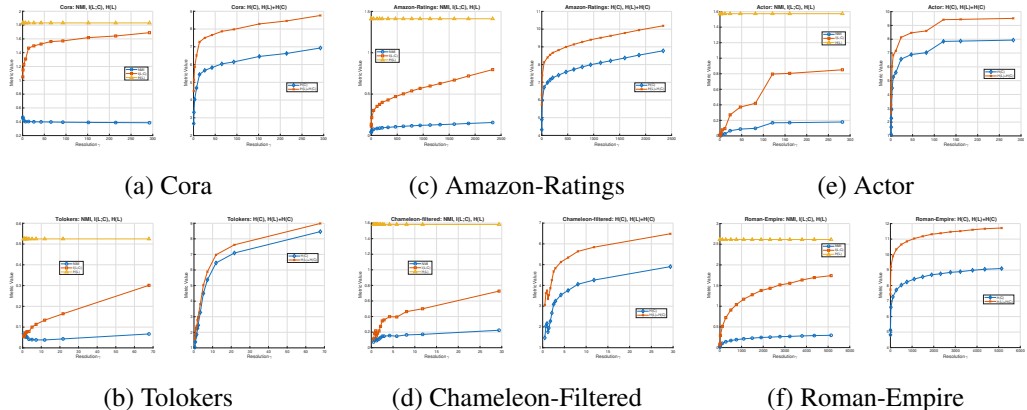

Figure 6: NMI, mutual information, and entropy dynamics across resolutions. high structural bias datasets (Cora and Tolokers, left); low structural bias datasets (Amazon and Chameleon-Filtered, middle); negative structural bias datasets (Actor and Roman-Empire, right).

- **Negative structural bias** (e.g., Actor, Squirrel-Filtered, Roman-Empire): Community structure is misaligned with labels, and finer communities introduce noisy or misleading locality, so refinement hurts. As $Q_{\min}$ is lowered and more fine-grained communities are added, performance deteriorates; see Figure 5 (top right and bottom right). In this regime, GCN typically underperforms the MLP baseline.

*Example (Cora).* For Cora (Fig. 5 top left; Table 7), each choice of $Q_{\min}$ selects a subset of resolution parameters: we include community features from all resolutions whose modularity satisfies $Q(\gamma) \geq Q_{\min}$. When $Q_{\min} \in \{1.0, 0.9\}$ no resolution meets the threshold, so ATLAS collapses to the feature-only MLP at $76.61\%$, below the GCN curve. At $Q_{\min} = 0.8$, two resolutions are added and accuracy rises to $79.93\%$; at $0.7$ a medium-resolution setting increases it to $83.66\%$; and at $0.6$ two finer resolutions push it to $86.50\%$. As $Q_{\min}$ is lowered further and more resolutions are added, the ATLAS curve eventually overtakes GCN, reaching its peak of $88.10\%$ at $Q_{\min} = 0.1$, where node features are augmented with a balanced mix of coarse, medium, and fine community features. Reducing $Q_{\min}$ to $0.0$ adds the most fragmented resolution and causes a slight drop, indicating diminishing returns from very fine community structure, which effectively acts as noise.

Figure 6 illustrates the refinement behavior predicted by our NMI theory. As resolution $\gamma$ increases, communities are refined and, as Lemmas 1–2 state, both $I(L;C)$ and $H(C)$ grow monotonically across all datasets. In contrast, Theorem 1 explains why NMI behaves differently across structural-

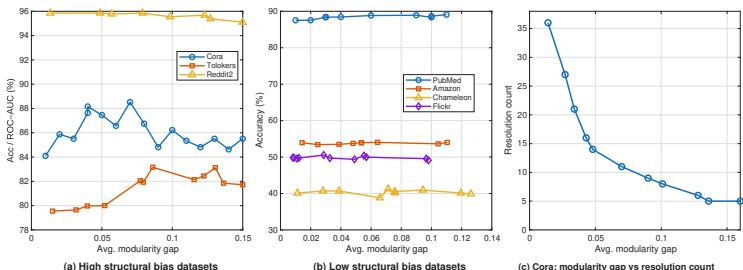

Figure 7: **(a–b)** Accuracy/ROC–AUC vs. average modularity gap on high- and low-structural-bias datasets. **(c)** Resolution count vs. modularity gap for Cora.

bias regimes: on high structural-bias graphs it forms a clear interior peak, on low structural-bias graphs it stays low and fairly flat because gains in $I(L; C)$ do not outpace the entropy increase, and on negative structural-bias graphs it rises slowly from near zero and saturates at a modest level.

## 6 ABLATION STUDY

**Effect of Modularity Gap and Resolution on Performance.** The behavior of refinement levels—how many Louvain resolution values are selected and how useful they are—is strongly shaped by the modularity gap. A small average modularity gap keeps many closely spaced resolutions, while a large gap leaves only a few widely separated ones. This creates a trade-off between having many redundant community partitions and having too few, overly coarse partitions. On high structural-bias graphs (Figure 7(a)), accuracy is lowest at the extremes of this trade-off. When the average gap is very small, ATLAS retains many nearly redundant partitions and the additional community features mostly inject noise, depressing accuracy. As the gap moves into a moderate region ($gap \approx 0.06 - 0.09$), the selected resolutions capture community structure at several distinct granularities and align better with the labels, so mutual information strengthens and accuracy improves. If the gap becomes too large ($gap \gtrsim 0.10$), only a handful of coarse resolutions remain; the community structure is too crude to fully exploit the available signal and performance falls again.

For low structural-bias graphs (Figure 7(b)), the accuracy curves are much flatter. In this setting, the community structure carries little information about the labels, so changing the modularity gap mostly just changes how many community resolutions are kept, without making them much more predictive. As a result, adding community features yields only modest gains over a feature-only MLP, and accuracy is only weakly affected by the choice of gap.

Figure 7(c) illustrates this behavior on Cora. For small gaps, many closely spaced resolutions are selected and their communities are highly overlapping, so the extra features are largely redundant and behave as noise, matching the low-accuracy regime in Figure 7(a). For large gaps, only a few coarse resolutions remain and the community information is too crude to capture label-relevant structure. The best performance occurs at intermediate gaps, where a small set of resolutions captures different levels of granularity, providing informative structural signal without redundancy.

## 7 CONCLUSION AND FUTURE WORK

We presented ATLAS, a community-augmented learning framework that enriches node features with multi-resolution Louvain embeddings and trains a compact MLP classifier. An adaptive resolution search, governed by $Q_{\min}$ and $\Delta Q$, selects a small set of informative resolutions, balancing coverage with cost. Across Q1 (homophily-regime benchmarks) and Q2 (large graphs), ATLAS attains competitive or superior accuracy relative to homophilic GNNs and heterophily-oriented models, while exhibiting fast, stable convergence and a favorable training footprint once preprocessing is complete.

In the future we aim to reduce preprocessing by improving resolution selection. A complementary direction is community-guided graph rewiring: using the discovered communities to propose sparse, label-aware edge edits that amplify useful intra-/inter-community signals and further improve accuracy.

**Ethics Statement.** We confirm that we have adhered to the ICLR Code of Ethics.

**Use of Generative AI.** We have used generative AI to polish the writing, and to check that the proofs of the theorem and lemma are correct and concise.

**Reproducibility Statement.** Our source code is available at `https://github.com/atlaspaper16/ATLAS`. This Github repository is created through an anonymous account, and thus does not violate the double-blind policy.

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

# 8 APPENDIX

**Compute environment.** All experiments were run on a server with $1\times$ NVIDIA A40 (45 GiB) GPU, 32 vCPUs, $2\times$ Intel Xeon Silver 4309Y @ 2.80 GHz, and 503 GiB RAM.
Software stack: Python 3.10.18; PyTorch 2.4.0+cu124 (CUDA 12.4); PyTorch Geometric 2.6.1.

## 8.1 DEFINITIONS AND TERMINOLOGY FOR COMMUNITY DETECTION

This subsection defines the modularity-based community detection terms that underpin our multi-resolution refinement.

**Modularity.** Given a partition $C = \{C_1, \ldots, C_K\}$ with node assignments $c_i \in \{1, \ldots, K\}$, *modularity* measures how much denser the intra-community connections are than expected under a degree-preserving null model:

$$Q = \frac{1}{2m} \sum_{i,j} \left( A_{ij} - \frac{k_i k_j}{2m} \right) \delta(c_i, c_j), \qquad (8)$$

where $A_{ij}$ is the adjacency matrix, $k_i$ is the degree of node $i$, $m = |E|$ is the number of edges, and $\delta(c_i, c_j) = 1$ if $c_i = c_j$ (else 0). Higher $Q$ indicates stronger community structure.

**Resolution parameter.** Louvain introduces a *resolution* $\gamma > 0$ to control granularity by reweighting the null-model term:

$$Q(\gamma) = \frac{1}{2m} \sum_{i,j} \left( A_{ij} - \gamma \frac{k_i k_j}{2m} \right) \delta(c_i, c_j). \tag{9}$$

Smaller $\gamma$ favors coarser partitions, while larger $\gamma$ typically yields finer (more, smaller) communities, producing a refinement hierarchy across $\gamma$. We denote the resulting partition and modularity by $C(\gamma)$ and $Q^{(\gamma)}$, with assignment vector $\mathbf{c}^{(\gamma)}$.

**Modularity gap.** For two consecutive tested resolutions $\gamma_1 < \gamma_2$, the *modularity gap* quantifies the change in community quality:

$$\Delta Q(\gamma_1, \gamma_2) = \left| Q^{(\gamma_2)} - Q^{(\gamma_1)} \right|. \tag{10}$$

Large gaps indicate rapid structural changes between scales and motivate inserting intermediate resolutions; small gaps suggest the refinement has stabilized.

## 8.2 THEORETICAL PROOFS

**Lemma. 1.** *Let $L$ be labels and $C$ a community partition. Let $C'$ be a refinement of $C$, i.e., $C' \preceq C$. Then $I(L; C') \geq I(L; C)$*

*Proof.* Let total number of elements be $n$. Then based on the definitions of $I(P, Q)$ in Section 2;

$$I(L; C') = \frac{1}{n} \sum_l \sum_{c'} n_{l,c'} \log\left( \frac{n\, n_{l,c'}}{n_l\, n_{c'}} \right), \qquad I(L; C) = \frac{1}{n} \sum_l \sum_c n_{l,c} \log\left( \frac{n\, n_{l,c}}{n_l\, n_c} \right),$$

where $n_l = \sum_{c'} n_{l,c'}$.

$$I(L; C') - I(L; C)$$
$$= \frac{1}{n} \sum_l \sum_{c'} n_{l,c'} \log\left( \frac{n\, n_{l,c'}}{n_l\, n_{c'}} \right) - \frac{1}{n} \sum_l \sum_c n_{l,c} \log\left( \frac{n\, n_{l,c}}{n_l\, n_c} \right)$$
$$= \frac{1}{n} \sum_c \sum_{c' \subseteq c} \sum_l \left[ n_{l,c'} \log\left( \frac{n\, n_{l,c'}}{n_l\, n_{c'}} \right) - n_{l,c'} \log\left( \frac{n\, n_{l,c}}{n_l\, n_c} \right) \right]$$
$$= \frac{1}{n} \sum_c \sum_{c' \subseteq c} \sum_l n_{l,c'} \log\left( \frac{n_{l,c'}\, n_c}{n_{l,c}\, n_{c'}} \right).$$

Since every $c' \subseteq of c$, therefore $\frac{n_{l,c'}}{n_{c'}} \geq \frac{n_{l,c}}{n_c}$. Thus the value in the log is positive, and $I(L; C') \geq I(L; C)$ $\qquad\square$

**Lemma. 2.** *Let $C$ a community partition. Let $C'$ be a refinement of $C$, i.e., $C' \preceq C$. Then $H(C') \geq H(C)$*

*Proof.* Let total size $n$. Based on the definition in Section 2

$$H(C) = -\sum_c \frac{n_c}{n} \log \frac{n_c}{n}, \qquad H(C') = -\sum_{c'} \frac{n_{c'}}{n} \log \frac{n_{c'}}{n}.$$

By grouping the $c'$ under their parent $c$:

$$H(C') - H(C) = -\sum_c \sum_{c' \subseteq c} \frac{n_{c'}}{n} \log \frac{n_{c'}}{n} + \sum_c \frac{n_c}{n} \log \frac{n_c}{n}$$

$$= \frac{1}{n} \sum_c \left[ -\sum_{c' \subseteq c} n_{c'} \log n_{c'} + n_c \log n_c \right].$$

Since $f(x) = -x \log x$ is a concave function and $c' \subseteq c$, therefore,

$$\sum_{c' \subseteq c} -\frac{n_{c'}}{n} \log \frac{n_{c'}}{n} \geq -\frac{n_c}{n} \log \frac{n_c}{n}.$$

Thus, $H(C') \geq H(C)$.

$\square$

**Theorem 1.** *Let $L$ be labels; $C$ a community partition. Let $C'$ be a refinement of $C$, i.e., $C' \preceq C$. Then NMI$(C'; L) >$ NMI$(C; L)$ if and only if $\frac{\Delta I}{\Delta H} > \frac{NMI(C;L)}{2}$; where $\Delta I = I(C'; L) - I(C; L)$ and $\Delta H = H(C'; L) - H(C; L)$*

*Proof.*

$$\text{NMI}(C; L) = \frac{2\, I(C; L)}{H(C) + H(L)}.$$

$$I := I(C; L), \qquad I' := I(C'; L), \qquad H := H(C), \qquad H' := H(C'), \qquad H_L := H(L).$$

Also

$$\Delta I := I' - I, \qquad \Delta H := H' - H.$$

Based on Lemma 1 and 2, $\Delta I \geq 0$ and $\Delta H \geq 0$. We do not consider edge case where $\Delta H = 0$. To show

$$\text{NMI}(C'; L) > \text{NMI}(C; L) \quad \Longleftrightarrow \quad \frac{\Delta I}{\Delta H} > \frac{\text{NMI}(C; L)}{2}.$$

$$\text{NMI}(C'; L) > \text{NMI}(C; L) \Longleftrightarrow \frac{2I'}{H' + H_L} > \frac{2I}{H + H_L}.$$

$$\frac{I'}{H' + H_L} > \frac{I}{H + H_L} \Longleftrightarrow I'(H + H_L) - I(H' + H_L) > 0.$$

Expand using $I' = I + \Delta I$ and $H' = H + \Delta H$:

$$(I + \Delta I)(H + H_L) - I(H + \Delta H + H_L) > 0.$$

Simplify terms (the $I(H + H_L)$ cancel):

$$\Delta I\, (H + H_L) - I\, \Delta H > 0.$$

Thus;

$$\Delta I\, (H + H_L) > I\, \Delta H \quad \Longleftrightarrow \quad \frac{\Delta I}{\Delta H} > \frac{I}{H + H_L}.$$

By definition $\text{NMI}(C; L) = \frac{2I}{H + H_L}$, so $\frac{I}{H + H_L} = \frac{\text{NMI}(C; L)}{2}$. Therefore

$$\text{NMI}(C'; L) > \text{NMI}(C; L) \quad \Longleftrightarrow \quad \frac{\Delta I}{\Delta H} > \frac{\text{NMI}(C; L)}{2}$$

$\square$

## 8.3 NMI ANALYSIS FOR DATASETS

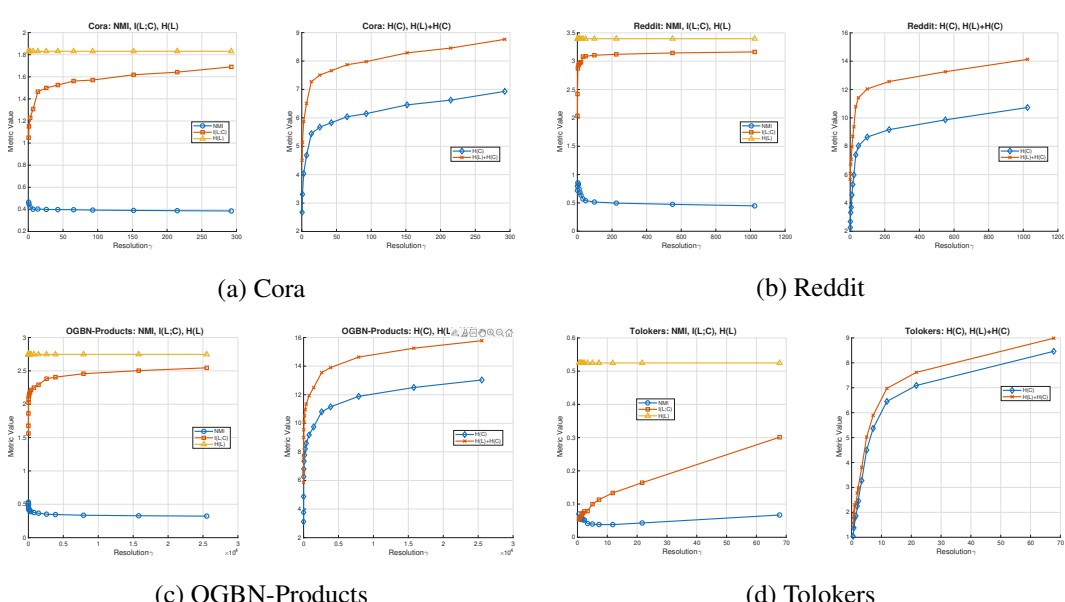

(a) Cora

(b) Reddit

(c) OGBN-Products

(d) Tolokers

Figure 8: High structural bias datasets: (a) Cora, (b) Reddit, (c) OGBN-Products, and (d) Tolokers. Each subfigure reports how $NMI$, $I(L; C)$, $H(L)$, $H(C)$, and $H(L)+H(C)$ vary as the resolution parameter $\gamma$ varies and yields community partitions of different granularity.

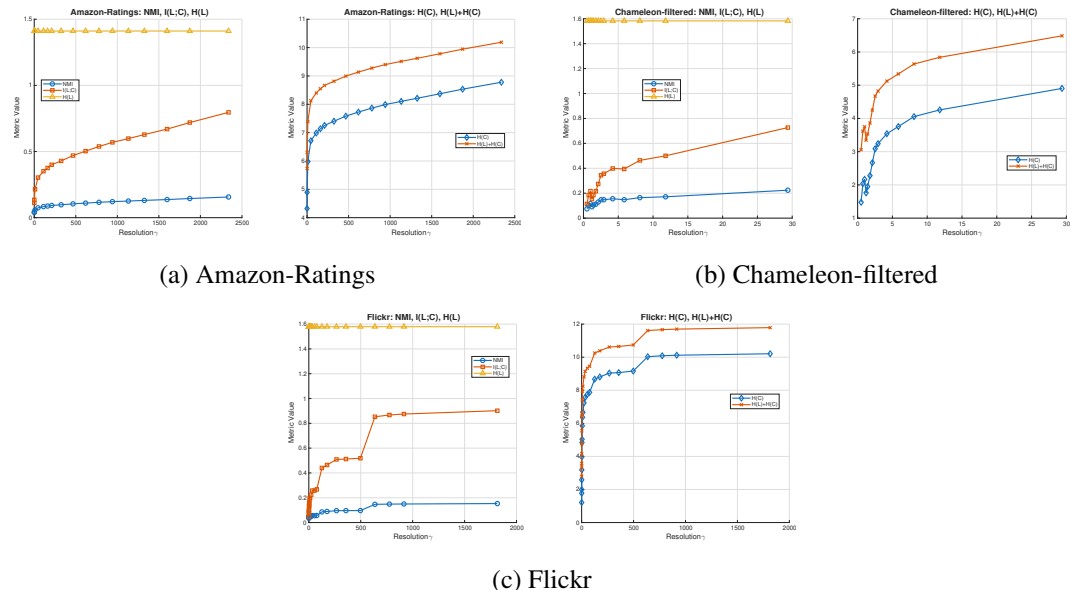

Figure 9: Low structural bias datasets: (a) Amazon-Ratings, (b) Chameleon-filtered, and (c) Flickr. Each subfigure illustrates how $NMI$, $I(L;C)$, $H(L)$, $H(C)$, and $H(L)+H(C)$ evolve as the resolution parameter $\gamma$ varies and yields community partitions of different granularity.

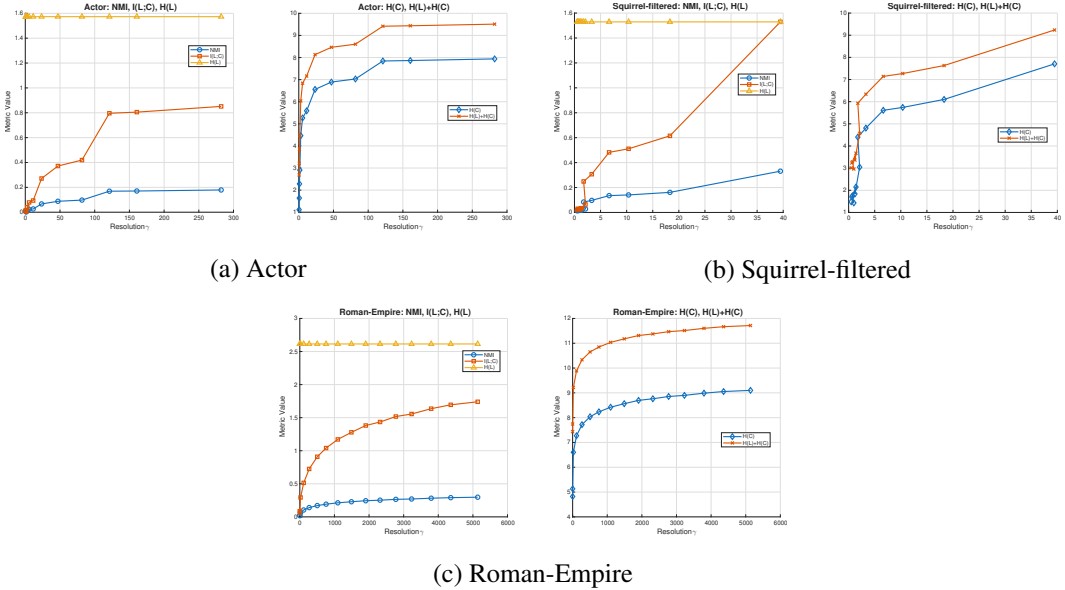

Figure 10: Negative structural bias datasets: (a) Actor, (b) Squirrel-filtered, and (c) Roman-Empire. Each subfigure reports how $NMI$, $I(L;C)$, $H(L)$, $H(C)$, and $H(L)+H(C)$ vary as the resolution parameter $\gamma$ varies and yields community partitions of different granularity.

## 8.4 ALGORITHMS

---

**Algorithm 1** Adaptive Resolution Search for Louvain

---

**Require:** graph $G$, minimum modularity $Q_{\min}$, maximum modularity gap $\Delta_{\max}$, gap_range= $[a, b]$
**Ensure:** resolutions, community_list

1: $\mathcal{C} \leftarrow \emptyset$; $Q \leftarrow \emptyset$
2: **for** $r \in \{0.5, 1.0\}$ **do**                  ▷ initial resolutions
3:      $(\mathcal{C}[r], Q[r]) \leftarrow \text{LOUVAIN}(G, r)$
4: **while** true **do**
5:      $L \leftarrow \text{SORTEDKEYS}(Q)$; $r_{\max} \leftarrow L[-1]$; $\tau \leftarrow Q[r_{\max}]$
6:      **if** $\tau \leq Q_{\min}$ **then**
7:          **break**
8:      new_r $\leftarrow$ None
9:      **for** consecutive $(r_1, r_2) \in L$ **do**
10:          **if** $|Q[r_2] - Q[r_1]| > \Delta_{\max}$ **then**
11:              new_r $\leftarrow (r_1 + r_2)/2$; **break**            ▷ interpolate
12:      **if** new_r= None **then**                ▷ extrapolate
13:          sample $\delta \sim \mathcal{U}[a, b]$; $Q^\star \leftarrow \tau - \delta$
14:          $s \leftarrow \text{ESTIMATESLOPE}(Q \text{ vs } r)$;
15:          new_r $\leftarrow r_{\max} + \dfrac{Q^\star - \tau}{s}$
16:      $(\mathcal{C}[\text{new\_r}], Q[\text{new\_r}]) \leftarrow \text{LOUVAIN}(G, \text{new\_r})$
17: resolutions $\leftarrow \{r \in \text{SORTEDKEYS}(Q): Q[r] \geq Q_{\min}\}$
18: community_list $\leftarrow [\mathcal{C}[r] \text{ for } r \in \text{resolutions}]$
19: **return** resolutions, community_list

---

**Algorithm 2** Community-Augmented Feature Projection for Node Classification

---

**Require:** Graph $G = (V, E)$, node features $\mathbf{X} \in \mathbb{R}^{n \times D}$, resolution set $\Gamma = \{\gamma_1, \ldots, \gamma_T\}$, projection dimension $d_c$
**Ensure:** Predicted label distribution $\hat{\mathbf{Y}} \in \mathbb{R}^{n \times C}$

1: Initialize empty list of embeddings $\mathcal{E}_{\text{emb}} \leftarrow [\,]$
2: **for** $\gamma \in \Gamma$ **do**
3:      Compute community assignment $\mathbf{c}^{(\gamma)} \in \mathbb{N}^n$
4:      One-hot encode $\mathbf{c}^{(\gamma)}$: $\mathbf{H}^{(\gamma)} \in \{0, 1\}^{n \times k_\gamma}$
5:      Project via trainable weights: $\mathbf{E}^{(\gamma)} \leftarrow \mathbf{H}^{(\gamma)} \mathbf{W}^{(\gamma)}$, where $\mathbf{W}^{(\gamma)} \in \mathbb{R}^{k_\gamma \times d_c}$
6:      Append $\mathbf{E}^{(\gamma)}$ to $\mathcal{E}_{\text{emb}}$
7: Concatenate all embeddings: $\mathbf{E} \leftarrow \text{Concat}(\mathcal{E}_{\text{emb}}) \in \mathbb{R}^{n \times (T \cdot d_c)}$
8: Concatenate with node features: $\mathbf{Z} \leftarrow [\mathbf{X} \,\|\, \mathbf{E}] \in \mathbb{R}^{n \times (D + T \cdot d_c)}$
9: Predict logits with MLP: $\mathbf{Y} \leftarrow f_\theta(\mathbf{Z}) \in \mathbb{R}^{n \times C}$
10: Apply softmax: $\hat{\mathbf{Y}} \leftarrow \text{softmax}(\mathbf{Y})$
11: **return** $\hat{\mathbf{Y}}$

---

## 8.5 COMPUTATION TIME ON LARGE GRAPHS

Table 5: Preprocessing (community detection), training, and inference times.

| Dataset | Preprocessing Time | Per-epoch Train Time | Inference Time |
|---|---|---|---|
| Reddit | $84.904 \pm 2.764$ | $0.143 \pm 0.002$ | $0.150 \pm 0.005$ |
| Flickr | $6.800 \pm 1.741$ | $0.241 \pm 0.005$ | $0.056 \pm 0.012$ |
| Yelp | $15.842 \pm 0.007$ | $2.670 \pm 0.007$ | $1.613 \pm 0.016$ |
| AmazonProducts | $72.270 \pm 1.409$ | $6.073 \pm 0.039$ | $3.056 \pm 0.019$ |

## 8.6 HYPERPARAMETER DETAILS

| Dataset | $Q_{\min}$ | $\Delta Q$ | Epochs | Batch | Hidden | Layers | Dropout | LR |
|---|---|---|---|---|---|---|---|---|
| Cora | 0.1 | 0.2 | 200 | 128 | 256 | 3 | 0.5 | 1e-4 |
| Pubmed | 0.7 | 0.1 | 300 | 8000 | 512 | 3 | 0.7 | 1e-4 |
| Tolokers | 0.3 | 0.1 | 2000 | 512 | 512 | 2 | 0.5 | 1e-4 |
| Squirrel-filtered | 0.61 | 0.05 | 60 | 512 | 512 | 3 | 0.5 | 5e-3 |
| Chameleon-filtered | 0.7 | 0.1 | 30 | 256 | 512 | 1 | 0.5 | 1e-3 |
| Amazon-ratings | 0.6 | 0.1 | 1500 | 512 | 512 | 3 | 0.5 | 1e-4 |
| Actor | 1.0 | 0.1 | 200 | 128 | 512 | 3 | 0.8 | 1e-4 |
| Roman-empire | 1.0 | 0.1 | 500 | 512 | 512 | 3 | 0.5 | 1e-4 |
| Flickr | 0.1 | 0.01 | 20 | 1024 | 256 | 2 | 0.7 | 1e-3 |
| Reddit | 0.3 | 0.3 | 1000 | 8000 | 512 | 3 | 0.5 | 1e-4 |
| Yelp | 1.0 | 0.1 | 300 | 32000 | 2048 | 5 | 0.5 | 5e-5 |
| AmazonProducts | 1.0 | 0.1 | 200 | 64000 | 2048 | 5 | 0.5 | 5e-5 |
| ogbn-products | 0.3 | 0.1 | 400 | 32000 | 512 | 3 | 0.5 | 1e-4 |

Table 6: Training hyperparameters by dataset. $Q_{\min}$ is the *minimum modularity* threshold and $\Delta Q$ is the *maximum modularity gap*.

Note. For `squirrel-filtered`, we explicitly use the community resolution 0.1. For `Tolokers`, we explicitly use the community resolution 0.5, 0.75, 1, 1.364. For `Pubmed`, we explicitly use the community resolution 0.5, 1, 1.956.

## 8.7 CORA: ACCURACY VS. MINIMUM MODULARITY THRESHOLD

Table 7 summarizes how relaxing the minimum modularity threshold $Q_{\min}$ on Cora changes both the community-derived features and the resulting accuracy. Each tuple $(Q, \text{Resolution}, \text{Communities})$ corresponds to a Louvain run at resolution $\gamma$: $Q$ is the modularity $Q(\gamma)$, and Communities is the number of communities $k_\gamma$ whose assignments $c^{(\gamma)}$ are one-hot encoded into $H^{(\gamma)}$ and projected to a dense embedding $E^{(\gamma)}$ that is concatenated into the multi-resolution community feature matrix (Algorithm 2). For a given $Q_{\min}$, the row lists the cumulative set of tuples with $Q \geq Q_{\min}$: blue tuples are newly activated at that threshold, while gray tuples persist from higher thresholds. When $Q_{\min} \geq 0.9$, no tuples qualify and the model reduces to the base MLP with accuracy 76.61%. As $Q_{\min}$ is lowered from 0.8 to 0.6, additional high-modularity, moderate-resolution community embeddings are added, and accuracy increases up to 86.50%. Further decreasing $Q_{\min}$ admits lower-modularity, finer resolutions with many more communities, leading to small fluctuations and a peak accuracy of 88.10% at $Q_{\min} = 0.1$, where a diverse mix of coarse-to-fine community features is used. Pushing $Q_{\min}$ to 0.0 adds one very fine tuple (768 communities), which slightly degrades performance to 86.32%, indicating that including too many extremely fine community features eventually injects noise.

Table 7: Cora: Cumulative ($Q$, Resolution, Communities) pairs included at each minimum modularity threshold $Q_{\min}$ (listed in run order), with accuracy. *Color coding:* pairs colored in blue are *newly added at that* $Q_{\min}$; pairs in gray were added at earlier thresholds and are carried over.

| Min Modularity $Q_{\min}$ | Pairs (Modularity, Resolution, Number of Communities) | Accuracy |
|:---:|:---:|:---:|
| 1.0 | — | 76.61 |
| 0.9 | — | 76.61 |
| 0.8 | (0.8526, 0.500, 90), (0.8120, 1.000, 103) | 79.93 |
| 0.7 | (0.8526, 0.500, 90), (0.8120, 1.000, 103), (0.7448, 2.606, 141) | 83.66 |
| 0.6 | (0.8526, 0.500, 90), (0.8120, 1.000, 103), (0.7448, 2.606, 141), (0.6841, 5.483, 170), (0.6006, 12.374, 298) | 86.50 |
| 0.5 | (0.8526, 0.500, 90), (0.8120, 1.000, 103), (0.7448, 2.606, 141), (0.6841, 5.483, 170), (0.6006, 12.374, 298), (0.5566, 20.068, 325) | 84.55 |
| 0.4 | (0.8526, 0.500, 90), (0.8120, 1.000, 103), (0.7448, 2.606, 141), (0.6841, 5.483, 170), (0.6006, 12.374, 298), (0.5566, 20.068, 325), (0.4909, 32.860, 373), (0.4231, 48.392, 430) | 86.15 |
| 0.3 | (0.8526, 0.500, 90), (0.8120, 1.000, 103), (0.7448, 2.606, 141), (0.6841, 5.483, 170), (0.6006, 12.374, 298), (0.5566, 20.068, 325), (0.4909, 32.860, 373), (0.3748, 63.924, 457), (0.4231, 48.392, 430) | 85.26 |
| 0.2 | (0.8526, 0.500, 90), (0.8120, 1.000, 103), (0.7448, 2.606, 141), (0.6841, 5.483, 170), (0.6006, 12.374, 298), (0.5566, 20.068, 325), (0.4909, 32.860, 373), (0.3748, 63.924, 457), (0.4231, 48.392, 430), (0.2784, 95.726, 541) | 82.59 |
| 0.1 | (0.8526, 0.500, 90), (0.8120, 1.000, 103), (0.7448, 2.606, 141), (0.6841, 5.483, 170), (0.6006, 12.374, 298), (0.5566, 20.068, 325), (0.4909, 32.860, 373), (0.3748, 63.924, 457), (0.4231, 48.392, 430), (0.2784, 95.726, 541), (0.1792, 136.430, 672) | 88.10 |
| 0.0 | (0.8526, 0.500, 90), (0.8120, 1.000, 103), (0.7448, 2.606, 141), (0.6841, 5.483, 170), (0.6006, 12.374, 298), (0.5566, 20.068, 325), (0.4909, 32.860, 373), (0.3748, 63.924, 457), (0.4231, 48.392, 430), (0.2784, 95.726, 541), (0.1792, 136.430, 672), (0.0958, 175.819, 768) | 86.32 |

## 8.8 MODELS FOR HOMOPHILIC GRAPHS

**GCN.** Applies a linear map followed by aggregation with the symmetrically normalized adjacency (after adding self-loops), corresponding to a first-order spectral/Chebyshev approximation (Kipf & Welling, 2017).

**GAT.** Learns attention coefficients over neighbors via masked self-attention and aggregates them with a softmax-weighted sum, enabling data-dependent receptive fields (Veličković et al., 2018).

**GraphSAGE.** Performs permutation-invariant neighbor aggregation (e.g., mean, max-pooling, LSTM) with fixed fan-out sampling per layer for scalable, inductive mini-batch training on large graphs (Hamilton et al., 2017).

## 8.9 MODELS FOR HETEROPHILIC GRAPHS

**H₂GCN.** Separates ego and neighbor embeddings, aggregates higher-order neighborhoods, and combines intermediate representations to improve robustness under heterophily (Zhu et al., 2020).

**LinkX.** Separately embeds node features and adjacency (structural) information with MLPs and concatenates them, capturing complementary attribute and topology signals that scale to non-homophilous graphs (Lim et al., 2021).

**GPR-GNN.** Learns signed polynomial (Generalized PageRank) propagation weights, adapting the filter to both homophilous and heterophilous label patterns and mitigating over-smoothing (Chien et al., 2021).

**FSGNN.** Applies soft selection over hop-wise aggregated features with "hop-normalization," effectively decoupling aggregation depth from message passing for a simple, shallow baseline that performs well under heterophily (Maurya et al., 2022).

**GloGNN.** Augments propagation with learnable correlations to global nodes (including signed coefficients), enabling long-range information flow and improved grouping on heterophilous graphs (Li et al., 2022).

**FAGCN.** Uses a self-gating, frequency-adaptive mechanism to balance low- and high-frequency components during message passing, improving robustness across homophily regimes (Bo et al., 2021).

**GBK-GNN.** Employs bi-kernel feature transformations with a gating mechanism to integrate homophily- and heterophily-sensitive signals within a single architecture (Du et al., 2022).

**JacobiConv.** Adopts an orthogonal Jacobi-polynomial spectral basis (often without nonlinearities) to learn flexible filters suited to varying graph signal densities, yielding strong performance on heterophilous data (Wang & Zhang, 2022).

**BernNet.** Learns general spectral graph filters using Bernstein polynomial approximation, enabling flexible control of low- and high-frequency components and strong performance under varying degrees of heterophily (He et al., 2021).

**ACM-GCN.** Uses high-pass filtering with adaptive channel mixing to combine low- and high-frequency components, yielding strong performance on heterophilic and mixed-regime graphs (Luan et al., 2022).

## 8.10 SAMPLING METHODS FOR SCALABLE GNNs

**GraphSAGE (node/neighbor sampling).** Samples a fixed fan-out of neighbors per layer and learns permutation-invariant aggregators, limiting the receptive field and enabling inductive, mini-batch training on large graphs (Hamilton et al., 2017).

**FastGCN (layer-wise node sampling).** Recasts graph convolution as an expectation over nodes and draws i.i.d. node sets at each layer via importance sampling, decoupling batch size from degree and reducing estimator variance (Chen et al., 2018b).

**S-GCN / VR-GCN (layer-wise with control variates).** Introduces control-variates using historical activations to stabilize gradients under small per-layer samples and achieve faster, provable convergence to the full-batch optimum (Chen et al., 2018a).

**ClusterGCN (subgraph/block sampling).** Partitions the graph and samples dense clusters as mini-batches, restricting propagation within blocks to boost edge coverage, cache locality, and memory efficiency at scale (Chiang et al., 2019).

**GraphSAINT (subgraph sampling with bias correction).** Constructs mini-batches by sampling subgraphs (node/edge/random-walk policies) and applies unbiased normalization to correct sampling bias, yielding strong accuracy–efficiency trade-offs on large graphs (Zeng et al., 2020).

## 8.11 DECOUPLING-BASED METHODS FOR SCALABLE GNNS

**SGC (linearized propagation).** Simplifies GCNs by collapsing multiple message-passing layers into a single $K$-step precomputation of $\mathbf{A}^K \mathbf{X}$, removing nonlinearities and train-time propagation. This reduces GNN training to logistic regression on pre-smoothed features, yielding strong scalability and fast inference (Wu et al., 2019a).

**SIGN (multi-hop feature precomputation).** Precomputes multiple graph-diffused feature channels (e.g., $\mathbf{A}^K \mathbf{X}$ for several $K$), and trains an MLP on the concatenated features. This decouples feature propagation from learning entirely, enabling embarrassingly parallel preprocessing and large-batch training (Rossi et al., 2020).

**SAGN (depth and scope decoupling).** Introduces a learnable gating mechanism over multiple precomputed hop-wise representations, allowing the model to adaptively weight short- and long-range information without stacking GNN layers. This stabilizes training under heterophily and yields strong performance with shallow architectures (Sun et al., 2021).

**GAMLP (self-ensemble on diffused features).** Builds an ensemble over diffused feature channels using attention and prediction consistency across hops. GAMLP reuses node features efficiently and achieves high accuracy with small models, while avoiding message passing during training and inference (Chien et al., 2022).

Together, these methods represent the broader "decoupling" paradigm—where propagation is performed once (or analytically) and training reduces to learning an MLP over fixed multi-hop representations—an approach systematically benchmarked and analyzed in large-scale settings by Zeng et al. (Zeng et al., 2022). ATLAS aligns with this propagation-free philosophy but differs fundamentally in how structural information is obtained: instead of precomputing $\mathbf{A}^k \mathbf{X}$, ATLAS extracts *multi-resolution community assignments* as topology-aware features, providing a complementary and scalable route to structural encoding.

## 8.12 DATASETS

We evaluate on two groups of benchmarks that stress complementary regimes.

**Large-scale graphs.** We use Flickr, Reddit, Yelp, AmazonProducts, and ogbn-products. Flickr/Yelp/AmazonProducts come from GraphSAINT; Reddit from GraphSAGE; ogbn-products from OGB (Zeng et al., 2020; Hamilton et al., 2017; Hu et al., 2020). Table 9 reports sizes, features, classes, and splits.

**Homophilous and heterophilous graphs.** We include Cora, PubMed, Actor, Chameleon-filtered, Squirrel-filtered, Amazon-ratings, Tolokers, and Roman-empire. For the filtered Wikipedia, Roman-empire, Amazon-ratings, and Tolokers datasets, we use the exact settings and splits of Platonov et al. (2023b); Cora, PubMed, and Actor follow standard preprocessing (Sen et al., 2008; Pei et al., 2020; Lim et al., 2021). Table 8 lists summary stats, edge homophily $h_e$, and metrics.

Table 8: Dataset statistics with edge homophily $h_e$ and evaluation metric ("Acc" for Accuracy, "ROC-AUC" for Area Under ROC).

| Dataset | Nodes | Edges | Avg. Degree | Feature | Classes | Train / Val / Test | $h_e$ | Metric |
|---|---|---|---|---|---|---|---|---|
| Cora | 2,708 | 5,429 | 4 | 1,433 | 7 (s) | 0.60 / 0.20 / 0.20 | 0.810 | Acc |
| PubMed | 19,717 | 44,324 | 5 | 500 | 3 (s) | 0.60 / 0.20 / 0.20 | 0.802 | Acc |
| Actor | 7,600 | 30,019 | 8 | 932 | 5 (s) | 0.60 / 0.20 / 0.20 | 0.216 | Acc |
| Squirrel-filtered | 2,223 | 65,718 | 59 | 2,089 | 5 (s) | 0.50 / 0.25 / 0.25 | 0.207 | Acc |
| Chameleon-filtered | 890 | 13,584 | 31 | 2,325 | 5 (s) | 0.50 / 0.25 / 0.25 | 0.236 | Acc |
| Amazon-ratings | 24,492 | 93,050 | 8 | 300 | 5 (s) | 0.50 / 0.25 / 0.25 | 0.380 | Acc |
| Tolokers | 11,758 | 519,000 | 88 | 10 | 2 (s) | 0.50 / 0.25 / 0.25 | 0.595 | ROC-AUC |
| Roman-empire | 22,662 | 32,927 | 3 | 300 | 18 (s) | 0.50 / 0.25 / 0.25 | 0.047 | Acc |

Table 9: Dataset statistics ("m" stands for **m**ulti-class classification, and "s" for **s**ingle-class.)

| Dataset | Nodes | Edges | Avg. Degree | Feature | Classes | Metric | Train / Val / Test |
|---|---|---|---|---|---|---|---|
| Flickr | 89,250 | 899,756 | 10 | 500 | 7 (s) | F1-micro | 0.50 / 0.25 / 0.25 |
| Reddit | 232,965 | 11,606,919 | 50 | 602 | 41 (s) | F1-micro | 0.66 / 0.10 / 0.24 |
| Yelp | 716,847 | 6,977,410 | 10 | 300 | 100 (m) | F1-micro | 0.75 / 0.10 / 0.15 |
| AmazonProducts | 1,598,960 | 132,169,734 | 83 | 200 | 107 (m) | F1-micro | 0.85 / 0.05 / 0.10 |
| ogbn-products | 2,449,029 | 61,859,140 | 50.5 | 100 | 47 (s) | Acc | 0.08 / 0.02 / 0.90 |

