# OpenReview forum: "ATLAS: Adaptive Topology -based Learning at Scale for Homophilic and Heterophilic Graphs"
_ICLR.cc/2026/Conference — Submitted to ICLR 2026_

### Official Review · Reviewer_cr2d · 2025-10-30

**Soundness:** 2
**Presentation:** 2
**Contribution:** 1
**Rating:** 2
**Confidence:** 4

**Summary:**

The paper proposes ATLAS, a topology-based learning method aiming to handle both homophilic and heterophilic graphs by incorporating community detection information into MLP features. The authors argue that this approach scales better than GNNs while maintaining comparable accuracy on various graphs.

**Strengths:**

The topic of homophily and heterophily learning on graphs with community detection is relevant and potentially interesting.

**Weaknesses:**

While the topic is relevant and potentially interesting, the paper suffers from several critical issues.

1. The literature review is insufficient, and the background and related work section is extremely weak. Key surveys and foundational works on heterophilic graph learning are missing, such as Zheng et al., “Graph Neural Networks for Graphs with Heterophily: A Survey” (arXiv:2202.07082, 2022), along with many subsequent heterophilic GNN studies.

2. The contribution lacks novelty, as the idea of combining community information with simple MLPs has already been explored (e.g., LinkX: Large Scale Learning on Non-Homophilous Graphs, NeurIPS 2021). The technical innovation over existing baselines is minimal.

3. Moreover, the experimental evaluation does not convincingly demonstrate superiority over recent heterophilic GNNs.

4. Lastly, the manuscript presentation is unpolished—the font and formatting differ noticeably from standard NeurIPS submissions, suggesting a lack of careful preparation.

**Questions:**

No more questions.

---

> ### Author Response · Authors · 2025-11-21
>
> We thank the reviewers for their detailed feedback. Below we restate the paper's core advances and then address each question in turn.
>
> ### Core Advances of ATLAS
>
> 1. **Topology-augmented learning without message passing.**
>    ATLAS constructs multi-resolution community features and inputs them to an MLP, enabling topology-aware learning without multiple rounds of message passing as required by GNNs.
>
> 2. **Theory linking refinement to accuracy.**
>    We show how refinement changes mutual information and entropy, predicting when coarse vs. fine structure helps in homophilic, weak-homophilic, and heterophilic regimes.
>
> 3. **Unified performance across regimes.**
>    By adaptively selecting refinement levels aligned with label coherence, ATLAS is competitive on both homophilic and heterophilic graphs.
>
> 4. **Scalability + Low inference cost (Adjacency free).**
>    Training is i.i.d. mini-batch MLP; inference is adjacency free with fixed augmented features.
>
> 5. **Bridging Frameworks.**
>    Unlike prior MLP-based models designed primarily for heterophilic graphs, ATLAS effectively supports both homophilic and heterophilic settings, thereby minimizing the accuracy gap traditionally observed between MLPs and GNNs. Moreover, its high inference efficiency positions ATLAS as a practical and scalable alternative to GNNs.
>
> 6. **Structural explainability.**
>    ATLAS provides explicit, interpretable community-based structural features—revealing which resolutions influence predictions—whereas standard GNNs lack this transparency due to opaque, entangled message-passing aggregation.
>
> ### Concerns and Responses
>
> **Concern 1: Related work insufficient / missing heterophily surveys & recent studies.**
> We substantially expanded Related Work to include key surveys (e.g., Zheng et al. 2022) and post‑2022 heterophily GNNs, plus a clearer taxonomy vs. propagation‑free methods.
>
> **Concern 2: “Community + MLP already explored (LinkX), so minimal innovation.”**
> LinkX **did not** use community information. While LinkX is also MLP‑based, ATLAS differs in several key aspects:
>
> 1. **Methodology.**
>    LinkX relies on the adjacency matrix, whereas ATLAS leverages multi‑resolution community information for feature construction.
>
> 2. **Homophily/Heterophily support.**
>    LinkX is designed specifically for heterophilic graphs, whereas ATLAS handles both homophilic and heterophilic graphs. For example, in our experiments, ATLAS outperforms LinkX on homophilic datasets like Cora (Table 1). By bridging this gap, ATLAS positions an MLP‑based framework as a practical rival to GNNs.
>
> 3. **Theoretical interpretation.**
>    LinkX does not provide an explanation of why their method works (or fails) on heterophilic graphs. In contrast, ATLAS offers a theoretical analysis (Section 2, Figure 5) showing how adaptive community refinement influences accuracy, modulated by graph modularity.
>
> **Concern 3: Experiments not convincingly stronger than recent heterophily GNNs.**
> We added ACM‑GCN and BernNet (Table 1, recent heterophily GNNs) and  SGC, SIGN, SAGN, GAMLP ( Table 2, large graph model) and ATLAS remains competitive. More importantly, ATLAS’s goal is a single unified, scalable alternative,  consistent across regimes, rather than a per‑regime specialized architecture.
>
> We also added an ablation (Sec. 4.1, Roman-Empire) showing the dataset is feature-dominated: neighborhood-feature amplification(concatenation) strengthens the *feature signal* , and when ATLAS uses the same amplification (ATLAS-NF), it closes to the FSGNN SOTA.
>
> **Updated**: New Chameleon-filtered results are added to Table 1, with revised hyperparameters.
> ATLAS/ATLAS-NF achieves the **best performance among all baselines** on **Tolokers, Chameleon-filtered, Amazon-Ratings**(Table 1), **Yelp, and Amazon-Products** (Table 2).
>
> **Concern 4: Presentation/format unpolished.** differ noticeably from standard **NeurIPS** submissions
>
> We adhere to the official **ICLR** LaTeX style and rechecked margins, fonts, and figure consistency per the author guide.

---

### Official Review · Reviewer_ZSd6 · 2025-11-01

**Soundness:** 2
**Presentation:** 2
**Contribution:** 2
**Rating:** 4
**Confidence:** 3

**Summary:**

The paper proposes ATLAS, a novel graph learning algorithm aimed at improving scalability and performance across both homophilic and heterophilic graphs. Instead of relying on neighborhood aggregation as in traditional GNNs, ATLAS extracts multi-level community features that encode topological structure and combines them with node features for classification using multilayer perceptrons (MLPs). This design eliminates costly message passing while preserving structural information, enabling efficient learning on large graphs. The authors also introduce a theoretical framework using normalized mutual information (NMI) to adaptively refine communities based on the degree of homophily, achieving strong interpretability and adaptability.

However, the experimental evaluation has notable weaknesses. On the OGBN-Products dataset, the set of comparison baselines is too limited and includes mostly outdated methods, making it difficult to assess ATLAS’s true competitiveness against modern scalable and heterophily-aware GNNs. Furthermore, while the method is conceptually sound, the problem it addresses—heterophilic rpoblem and graph scale—has been explored in prior research, reducing its overall novelty.

**Strengths:**

This work is well-structured and clearly written. The paper effectively connects theoretical analysis to practical implementation. This paper considers two important issues in the GNN, heterophilic problem and graph scale.

**Weaknesses:**

The theory part is hard to follow, and how this part helps the methododolegy is not clear. The experimental part needs more recent baslines.

**Questions:**

From my perspective, these baselines are before 2023, is that possible to compare with other recent works?

---

> ### Author Response · Authors · 2025-11-21
>
> We thank the reviewers for their detailed feedback. Below we restate the paper's core advances and then address each question in turn.
>
> ### Core Advances of ATLAS
>
> 1. **Topology-augmented learning without message passing.**
>    ATLAS constructs multi-resolution community features and inputs them to an MLP, enabling topology-aware learning without multiple rounds of message passing as required by GNNs.
>
> 2. **Theory linking refinement to accuracy.**
>    We show how refinement changes mutual information and entropy, predicting when coarse vs. fine structure helps in homophilic, weak-homophilic, and heterophilic regimes.
>
> 3. **Unified performance across regimes.**
>    By adaptively selecting refinement levels aligned with label coherence, ATLAS is competitive on both homophilic and heterophilic graphs.
>
> 4. **Scalability + Low inference cost (Adjacency free).**
>    Training is i.i.d. mini-batch MLP; inference is adjacency free with fixed augmented features.
>
> 5. **Bridging Frameworks.**
>    Unlike prior MLP-based models designed primarily for heterophilic graphs, ATLAS effectively supports both homophilic and heterophilic settings, thereby minimizing the accuracy gap traditionally observed between MLPs and GNNs. Moreover, its high inference efficiency positions ATLAS as a practical and scalable alternative to GNNs.
>
> 6. **Structural explainability.**
>   ATLAS provides explicit, interpretable community-based structural features—revealing which resolutions influence predictions—whereas standard GNNs lack this transparency due to opaque, entangled message-passing aggregation.
>
> ### Concerns and Responses
>
> **Concern 1: Theory hard to follow / unclear linkage to method.**
> We have added a bridge in the first paragraph of Section 3 (methodology), to link the theory to the method. We have further demonstrated in Figure 6, how the NMI curves follow the prediction outlined in our theory.
>
> **Concern 2: Baselines too old on OGBN-Products / missing recent scalable or heterophily methods.**
> We expanded baselines in two directions:
>
> - **Heterophily-aware GNNs:** Added ACM-GCN and BernNet.  ( Table 1 )
> - **Modern large-graph decoupling baselines:** Added SGC, SIGN, SAGN, GAMLP ( Table 2, Table 3 )following large-graph benchmarking practice in [1].
>
> These baselines are strong on OGBN-Products yet fail on heterophilic graphs, while ATLAS stays robust across regimes.
>
> We also checked the public OGBN-Products leaderboard [2]:  top entries largely rely on external data (“Ext. data: Yes”) and multi-stage pipelines, so they are not directly comparable to our single-model, official-features setting.
>
> We therefore benchmark against strong, reproducible well-accepted  single-model baselines rather than external-data or ensemble models.
>
> We also added an ablation (Sec. 4.1, Roman-Empire) showing the dataset is feature-dominated: neighborhood-feature amplification(concatenation) strengthens the *feature signal* , and when ATLAS uses the same amplification (ATLAS-NF), it closes to the FSGNN SOTA.
>
> **Updated**: New Chameleon-filtered results are added to Table 1, with revised hyperparameters.
> ATLAS/ATLAS-NF achieves the **best performance among all baselines** on **Tolokers, Chameleon-filtered, Amazon-Ratings**(Table 1), **Yelp, and Amazon-Products** (Table 2).
>
> [[1] Dataset and Benchmark Track Paper “A Comprehensive Study on Large Scale Graph Training: Benchmarking and Rethinking”](https://openreview.net/forum?id=2QrFr_U782Z)
>
> [[2] OGBN-Products Leaderboard (Node Property Prediction) — SNAP/OGB](https://snap-stanford.github.io/ogb-web/docs/leader_nodeprop/)
>
> **Concern 3: “Problems explored before → reduced novelty.”**
>
> Prior work explores pieces of our target space **in isolation**:
>
> 1. **Homophily via message passing.**
>    Standard message-passing GNNs mainly target homophilic graphs, where multi-hop aggregation works best.
>
> 2. **Heterophily within message passing**
>    Existing GNN variants address heterophily by modifying aggregation or propagation, but remain tied to multi-hop message passing.
> 3. **Scalability-aware designs for one regime**
>
>    Methods that scale focus on either homophilic or heterophilic graphs, but rarely both
> 4. **Adjacency-aware MLPs with flat topology signals.**
>    Methods such as LinkX encode topology only as a **fixed, single-scale adjacency feature**, lacking any **reasoning or explainability**.
>
> 5. **Community-aware methods at a single resolution**
>    Prior community-based approaches rely on **one partition level**
>
> **ATLAS is the first to combine**:
> - **No message passing**,
> - **Inference efficiency**,
> - **Adaptive multi-resolution community signals**
> - **A theory-backed process**   , and
> - **Structural explainability** through explicit community features—an ability standard GNNs lack due to opaque message-passing aggregation.
>
> Together, these components bridge the accuracy gap between MLPs and GNNs across homophilic and heterophilic graphs and position MLPs as practical rivals to GNNs.

---

### Official Review · Reviewer_gHTe · 2025-11-03

**Soundness:** 2
**Presentation:** 2
**Contribution:** 2
**Rating:** 4
**Confidence:** 4

**Summary:**

This paper proposes a simple yet powerful framework to overcome the scalability and heterophily limitations of traditional Graph Neural Networks. The method constructs multi-resolution community features using adaptive community refinement guided by Normalized Mutual Information (NMI), concatenates them with node features, and feeds them into an MLP—eliminating neighborhood aggregation and enabling adjacency-free inference.

**Strengths:**

The graph categorization based on structural bias is interesting.

**Weaknesses:**

See below

**Questions:**

1. "Accurate classification requires two orthogonal pieces of information–(i) the features at each node, and (ii) the connections between the node and its neighbors." These two components are not necessarily orthogonal to each other, and sometimes you need to disentangle their relation to investigate their impacts on GNN performance [1].

2. Section 3 is not well developed and need to be polished. It starts with some notations that are not properly introduced before (e.g. three hyperparameters, modularities), which is confusing. Also, it's better to emphasize its connection to the previous sections and the main storyline of this paper.

3. Is the community assignment algorithm the main contribution of your paper? How does it compare with other existing algorithms?

4. How does your "multi-resolution community" address the heterophily problem? You can provide some insights if there is no theoretical evidence.

5. Does the partitions of communities be aware of the label distribution? Does your proposed method add more hyperparameters to tune?

6. In table 2, you hightlight all results of your proposed method, however it is not the best among the baselines in all tasks.

7. You should introduce the definition of high/low/negative structural bias before using them.

8. Missing comparison with some baseline models, e.g. ACM-GCN and Bernnet [2,3]. Ablation study on the impact of different resolutions of communities is needed.


[1] What is missing for graph homophily? disentangling graph homophily for graph neural networks. Advances in Neural Information Processing Systems, 37, 68406-68452.

[2] Revisiting heterophily for graph neural networks. Advances in neural information processing systems. 2022 Dec 6;35:1362-75.

[3] Bernnet: Learning arbitrary graph spectral filters via bernstein approximation. Advances in neural information processing systems, 34, 14239-14251.

---

> ### Author Response · Authors · 2025-11-21
>
> We thank the reviewers for their feedback. Below we restate the paper's core advances and then address each question in turn.
>
> ### Core Advances of ATLAS
>
> 1. **Topology-augmented learning without message passing.**
>    ATLAS constructs multi-resolution community features and feeds them into an MLP, enabling topology-aware learning without repeated message passing in GNNs.
>
> 2. **Theory linking refinement to accuracy.**
>    We characterize how refinement changes mutual information and entropy, which predicts when coarse vs. fine structure helps in *homophilic*, *weak-homophilic*, and *heterophilic* regimes.
>
> 3. **Unified performance across regimes.**
>    By selecting refinement levels aligned with label coherence, ATLAS is competitive on  homophilic and heterophilic graphs.
>
> 4. **Scalability with low inference cost.**
>    Training uses i.i.d. mini-batch MLP optimization; inference is **adjacency-free** because augmented features are fixed after preprocessing.
>
> 5. **Bridging frameworks.**
>    Unlike prior MLP-based methods tuned mainly for heterophily, ATLAS supports both homophily and heterophily, reducing the MLP–GNN accuracy gap, keeping  inference efficiency.
>
> 6. **Structural explainability.**
>    ATLAS provides  interpretable community-based structural features—revealing which resolutions influence predictions—whereas GNNs lack this transparency due to opaque, entangled message-passing aggregation.
>
> ### Point-by-Point Responses
>
> **Q1. Complementary vs. orthogonal information.**
> Changed to **two complementary sources of information**
>
> **Q2. Section 3 clarity and missing notation.**
> In the revision, we:
> 1. briefly defined **modularity**, **resolution parameter**, and **modularity gap** in Section 3;
> 2. added a brief bridge at the start of Section 3 connecting the MI–entropy tradeoff (Section 2) to the adaptive refinement procedure; and
> 3. provided pointers to Appendix 8.1 for formal definitions.
>
> **Q3. Is community assignment the main contribution? Comparison to other algorithms.**
> Our contribution is twofold: (i) **novel graph learning algorithm**—a propagation-free, community-augmented MLP that unifies multi-resolution community features (ii) a **theory–algorithm link via an NMI-based structural-bias analysis**, to our knowledge, is not made explicit in prior models. We compare against strong homophilic, heterophilic, and large-scale GNN baselines.
>
> **Q4. How multi-resolution communities help in heterophily; linkage to theory.**
> ATLAS handles homophily/heterophily by matching structural information to the **scale where labels are most coherent**.
> In heterophilic graphs, 1-hop neighborhoods are often misleading, while broader structural groupings may align better with labels.
> Multi-resolution communities provide partitions from coarse to fine, so the model can exploit the level that best captures label coherence without relying on fragile local signals.
>
> The **minimum-modularity threshold** controls the structural bias:
> - **Higher thresholds** retain coarser communities, typically benefiting heterophilic graphs.
> - **Lower thresholds** admit finer communities, typically benefiting homophilic graphs.
>
> As shown in Fig. 5 of the revision, varying this threshold yields predictable accuracy changes across regimes.
> Moreover, Fig. 6, 8, 9, 10  demonstrates that NMI curves follow the trends predicted by MI–entropy refinement theory.
>
> **Q5. Are partitions label-aware? Do they add many hyperparameters?**
> Partitions are computed **purely from topology**; labels are never used to form communities.
> The only added hyperparameters are the refinement controls $(Q_{\min}, \Delta Q)$, and we include sensitivity/ablation results showing stable and predictable behavior.
>
> **Q6. Table highlighting vs. not best everywhere.**
> We corrected the formatting so that revised Tables 1–2 highlight best-in-column results.
> Our claim is **consistency across regimes**, not per-dataset SOTA on every benchmark.
> We also added an ablation (Sec. 4.1, Roman-Empire) showing the dataset is feature-dominated: neighborhood-feature amplification(concatenation) strengthens the *feature signal* , and when ATLAS uses the same amplification (ATLAS-NF), it closes to the FSGNN SOTA.
>
> **Updated**: New Chameleon-filtered results are added to Table 1, with revised hyperparameters.
> ATLAS/ATLAS-NF achieves the **best performance among all baselines** on **Tolokers, Chameleon-filtered, Amazon-Ratings**(Table 1), **Yelp, and Amazon-Products** (Table 2).
>
> **Q7. Define high/low/negative structural bias earlier.**
> We now introduce **high, low, negative structural bias** in Sec. 5 immediately before first use, and reinforce these definitions in the captions of Fig. 5 and Fig. 6.
>
> **Q8. Missing ACM-GCN / BernNet; need resolution ablation.**
> We added **ACM-GCN** and **BernNet** to the main benchmark suite (Table 1).
> We expanded the ablations to isolate the role of **resolution count** and **modularity gap** (Sec. 6, Fig. 7).

---

### Meta-Review · Area_Chair_k1oE · 2025-12-23

**Summary:**

The reviewers have raised the following main concerns:

(1) Missing recent benchmarks

(2) Theoretical gap: the relation between the multi-resolution community and the heterophily problem

(3) The proposed model does not achieve SOTA performance.

(4) The paper lacks novelty as compared with earlier work, such as LinkX.

(5) Certain parts of the paper require restructuring and polishing.

**Reviewer Concerns:**

In the revised manuscript, the authors have improved the overall structure of the paper in response to the reviewers’ comments and have included additional numerical studies to demonstrate the proposed model. However, some concerns remain unresolved.

While the authors have added benchmarks such as ACM-GCN and BernNet, these methods are not among the most recent developments. Although newer models do not necessarily guarantee better performance, the evaluation could be further strengthened by considering recent directions, such as diffusion-based approaches derived from differential-equation–governed dynamics. Additionally, the proposed model does not yet demonstrate a clear performance advantage over existing methods, including the newly added ACM-GCN. Clarifying the scenarios in which the proposed approach is most effective, or providing additional empirical or theoretical insights, could help better highlight its contributions.

In my view, the concerns regarding the theoretical gap are well-founded. The connection between the analysis in Section 2 and the empirical performance of the proposed model is not yet clearly articulated. In particular, for heterophilic graphs, it is not immediately evident how a community-based structural feature extraction can yield discriminative information for node classification, since a $k$-hop (larger $k$) community is likely to contain a mixture of nodes from different classes. The authors are encouraged to provide a more in-depth theoretical discussion to clarify this relationship and to better motivate the effectiveness of the proposed approach.

Regarding novelty, at a high level, the proposed architecture bears a strong resemblance to LinkX, with the primary differences lying in the design of specific sub-components. The authors are therefore encouraged to more clearly articulate why these design choices are essential to achieving superior performance, ideally by providing both theoretical justification and empirical evidence.

**Reviewer Scores:**

None of the reviewers has actively participated in the discussion. It therefore remains unclear whether they will revise their scores, as their concerns appear to be only partially addressed in the rebuttal.

---

### Decision · Program_Chairs · 2026-01-26

Reject